# Furfuryl Alcohol and Lactic Acid Blends: Homo- or Co-Polymerization?

**DOI:** 10.3390/polym11101533

**Published:** 2019-09-20

**Authors:** Lukas Sommerauer, Jakub Grzybek, Michael S. Elsaesser, Artur Benisek, Thomas Sepperer, Edgar Dachs, Nicola Hüsing, Alexander Petutschnigg, Gianluca Tondi

**Affiliations:** 1Forest Products Technology & Timber Constructions Department, Salzburg University of Applied Sciences, Marktstraße 136a, 5431 Kuchl, Austria; lsommerauer.htw-m2017@fh-salzburg.ac.at (L.S.); jgrzybek.htw-m2018@fh-salzburg.ac.at (J.G.); thomas.sepperer@fh-salzburg.ac.at (T.S.); alexander.petutschnigg@fh-salzburg.ac.at (A.P.); 2Department of Chemistry and Physics of Materials, Paris-Lodron-University Salzburg, Jakob-Haringer-Strasse 2A, 5020 Salzburg, Austria; Michael.Elsaesser@sbg.ac.at (M.S.E.); artur.benisek@sbg.ac.at (A.B.); edgar.dachs@sbg.ac.at (E.D.); nicola.huesing@sbg.ac.at (N.H.); 3Salzburg Center for Smart Materials, Jakob-Haringer-Strasse 2A, 5020 Salzburg, Austria; 4Department of Land, Environment, Agriculture & Forestry, University of Padua, Via dell´Universitá 16, 35020 Legnaro, Italy

**Keywords:** Carbohydrate derivatives, sustainable macromolecules, bio-based materials, bio-resources, green resins, biopolymers

## Abstract

Furfuryl alcohol (FA) and lactic acid (LA) are two of the most interesting biomolecules, easily obtainable from sugars and hence extremely attractive for green chemistry solutions. These substances undergo homopolymerization and they have been rarely considered for copolymerization. Typically, FA homopolymerizes exothermically in an acid environment producing inhomogeneous porous materials, but recent studies have shown that this reaction can be controlled and therefore we have implemented this process to trigger the copolymerization with LA. The mechanical tests have shown that the blend containing small amount of FA were rigid and the fracture showed patterns more similar to the one of neat polyfurfuryl alcohol (PFA). This LA-rich blend exhibited higher chloroform and water resistances, while thermal analyses (TG and DSC) also indicated a higher furanic character than expected. These observations suggested an intimate interconnection between precursors which was highlighted by the presence of a small band in the ester region of the solid state ^13^C–NMR, even if the FT-IR did not evidence any new signal. These studies show that these bioplastics are basically constituted of PLA and PFA homopolymers with some small portion of covalent bonds between the two moieties.

## 1. Introduction

Carbohydrates are the most abundant class of molecules produced by living processes. They are obtained directly and indirectly through photosynthesis and therefore they are the most attractive compounds for replacing oil-based precursors [1]. Two of the most interesting molecules derived from carbohydrates are furfuryl alcohol (FA) and lactic acid (LA) because they can generate macromolecules through homo-polymerization to polyfurfuryl alcohol (PFA) and polylactic acid (PLA), respectively [2,3].

Furfuryl alcohol can be synthesized from sugar derivatives such as hemicelluloses and starch [4,5,6]. It is possible to polymerize FA in acid environment to polyfurfuryl alcohol (PFA) which is a thermosetting macromolecule with extremely high dimensional stability at high temperature (up to 340 °C) [7] and therefore used also in the foundry industry [8,9]. In recent years, FA polymers have also gained interest due to their important role in copolymerizing phenolic bioresources [10,11] and in the production of innovative lightweight materials such as tannin and lignin foams [12,13,14,15]. Polyfurfuryl alcohol is a very well-studied polymer. Detailed investigations concerning its thermosetting nature showed that a major part consists of linear polycondensation products, while a minor part is composed of networks that are produced from ring opening and Diels–Alder rearrangements [2,16,17,18,19]. Polyfurfuryl alcohol is typically produced through a highly exothermic, acid catalyzed polymerization reaction of furfuryl alcohol which is difficult to control and produces high porous inhomogeneous random structures [20,21]. Only very recently, a three-steps method to produce this polymer in a controlled manner was developed [22] and this breakthrough finally allows to consider FA as comonomer for new bio-materials [23,24].

Lactic acid is also a bio-derived molecule that can be obtained by bacterial fermentation of sugars [25,26,27]. The possibility of lactic acid to undergo polymerization to polylactic acid (PLA) is known since very long time [28], but only recently this product has gained greater interest because of its extensive use in the field of mulch films [29,30], biodegradable plastics [31,32], wood preservation [33,34], biomedicine [35] and 3D printing [36,37,38]; copolymers PLA with PEG are also well known for their applications as drug carriers [39]. Its homopolymerization occurs via polycondensation and can be catalyzed by various catalysts [3,40,41]. These catalysts increase the complexity of the system and therefore a catalyst-free reaction, would promote the synthesis of completely biogenic, sustainable polymers.

The idea of the present study is to run the polymerization of the two bio-precursors simultaneously in order to facilitate the copolymerization that may occur according to several pathways through lactic acid esterification with: (i) furfuryl alcohol before both precursors start their homopolymerization; (ii) the enolic form of the ring-opened fraction of PFA; (iii) the levulinic acid produced as intermediate before chain termination [42].

In this context we have considered the possibility to synthetize FA and LA copolymers in different relative proportions (100% PFA, 75%-25%, 50%-50%, 25%-75% and 100% PLA) applying the controlled polymerization conditions. The produced bioplastics were characterized applying SEM imaging, solubility tests, TGA, DSC, FT-IR and solid state ^13^C–NMR spectroscopy, as well as mechanical tests.

## 2. Materials and Methods

### 2.1. Materials

Furfuryl alcohol was kindly supplied by TransFuran Chemicals (Geel, Belgium). L (+) Lactic acid (80%) was purchased by Roth (Karlsruhe, Germany). A 32% sulfuric acid (Merck, Kennborough, NJ, USA) solution was used exclusively to prepare PFA.

### 2.2. Synthesis of PFA–PLA Polymers

The polymers were produced by filling silicone made molds with the formulation reported in Table 1 and following a controlled curing process which consisted of four phases: room temperature for 4 h, then heating to 50 °C for 22 h, further increase the temperature to 100 °C for 285 h followed by a final hardening phase at 150 °C for 24 h. The process is similar to the one presented by Sharib et al. [22], but extended in phase 3 and 4 to obtain the polymerization completion of lactic acid.

The solid polymers obtained were removed from the silicone container and stabilized at 20 °C and 65% relative humidity for 1 day and then tested for their bending resistance. The broken samples were observed under SEM and successively grinded into fine powder. The latter were tested for their solubility in chloroform and water. TGA, DSC, FT-IR and solid state ^13^C–NMR spectroscopy analyses were also performed on the water leached samples.

### 2.3. Bending Test

The samples prepared for the mechanical bending tests and for the SEM analysis were produced in a silicone mold with 80 × 25 mm^2^ dimension and final thickness between 3 and 6 mm. Three-point bending tests were performed using a Zwick-Roell Z250 Universal testing machine (Zwick-Roell, Ulm, Germany) equipped with a 1kN load cell. Crosshead speed was set to 2 mm/min, distance between support and loading nose was 56 mm.

### 2.4. Scanning Electron Microscopy (SEM)

After the bending test, one specimen for each formulation was investigated in its fracture profile using a Zeiss Ultra Plus field emission scanning electron microscope (SEM) equipped with an annular backscatter electron detector (Carl Zeiss AG, Oberkochen, Germany). Acceleration voltage was set to 5 kV and working distance was adjusted between 4 mm and 6 mm. Prior to the imaging, the samples were coated with a thin layer of gold using sputter coater with a current of 40 mA and coating time of 60 sec.

### 2.5. Solubility Tests

1.0 gram of the solid polymer powder (*W*_i_) was poured in 100 mL deionized water or chloroform under continuous magnetic stirring. After 1 h, the suspension was filtered with dry filter paper (*W_0_*). The filter was then dried in an oven T = 103 °C until constant weight (*W*_t_). The final dry weight *W*_f_
*= W*_t_−*W*_0_. The solubility (*S*) expressed in percentage were calculated according to the following formula: $$S=\left(1-\frac{W_{f}}{W_{i}}\right)\cdot 100\;\left[\%\right]$$

### 2.6. Thermogravimetric Analysis (TG)

For each formulation 20–30 mg of the pulverized material were deposited in an aluminum oxide crucible and analysed with a Netzsch STA 449 F3 Jupiter TG analyzer (Netzsch, Selb, Germany). The experiments were conducted between 20 and 1000 °C in synthetic air atmosphere with a rate of heating of 10 °C/min.

### 2.7. Differential Scanning Calorimetry (DSC)

The heat capacity between 0 and 200 °C was measured on the five powders using a power compensated Perkin Elmer Diamond DSC^®^ (Perkin-Elmer, Waltham, MA, USA) on samples weighing 20–22 mg. Prior to the measurements, the samples were heated at 150 °C for 15 min to drive off any water or adsorbed gases. The DSC measurements were performed under a flow of Ar gas, with the calorimeter block kept at -20 °C using a Perkin Elmer Intracooler. Each measurement consisted of a blank, a calibration and a sample run. The heat flow data (difference in heating power between the sample and the reference chamber) were collected using a temperature scan (heating rate of 10 °C/min) and isothermal periods of 3 min before and after the temperature scan. The heat flow versus temperature data from the sample runs were shifted and rotated until the data of the isothermal periods agreed with those of the blank run [43]. The data from the blank run were then subtracted from those of the sample run to give the net heat flow of the sample. For calculating the heat capacity, the net heat flow data were finally divided by the heating rate and the mass of the sample. Each sample run was corrected against a calibration run using a synthetic single crystal of corundum (31.764 mg), whose heat capacities were taken from the National Bureau of Standards Certificate [44] from 10 to 1950 K. The uncertainty of the DSC heat capacity data was determined to be better than 0.6%.

### 2.8. FT-IR Spectroscopy

The cured formulations were analyzed through ATR FT-IR spectroscopy with a PerkinElmer Frontier spectrometer (Perkin-Elmer, Waltham, MA, USA) provided by an ATR Miracle diamond crystal with the same acquisition parameters of previous studies [45]. Shortly, the powders were laid on the diamond eye (*d* = 1.8 mm) of the ATR equipment and the contact was ensured by tightly screwing the clamp. Each sample was scanned registering the spectrum with 32 scans with a resolution of 4 cm^−1^ in the wavenumber range between 4000 and 600 cm^−1^. Every polymer was scanned three times and the average of these spectra after baseline correction and area normalization was obtained with the Unscrambler (CAMO, Oslo, Norway) software in the fingerprint region between 1800 and 600 cm^−1^.

### 2.9. Solid State ^13^C–NMR Spectroscopy

The five powders were measured with a Bruker Avance NEO 500 wide bore system (Bruker BioSpin, Rheinstetten, Germany) solid state ^13^C–NMR spectrometer at the NMR center of the faculty of chemistry University of Vienna. A 4 mm triple resonance magic angle spinning (MAS) probe was used with a resonance frequency for ^13^C of 125.78 MHz, the MAS rotor spinning was set to 14 kHz. Cross polarization (CP) was achieved by a ramped contact pulse with a contact time of 2 ms. During acquisition ^1^H was high power decoupled using SPINAL (Bruker BioSpin, Rheinstetten, Germany) with 64 phase permutations. The chemical shifts for ^13^C are reported in ppm and are referenced external to adamantane by setting the low field signal to 38.48 ppm. The data elaboration was done with the software Top-spin 4.0.6 (Bruker, Billerica, MA, USA) while the calculations of the theoretical chemical shifts were done with the software NMR-Predict developed by the University of Lausanne (Luc Patiny) and the University of del Valle (Julien Wist) [46,47,48].

## 3. Results and Discussion

### 3.1. Preparation Method

Five samples series covering furfuryl alcohol to lactic acid (F/L) ratios from 100:0, 75:25, 50:50, 25:75 and 0:100 (Table 1) were prepared and their physical and chemical properties were investigated in detail.

The curing process of the five evaluated formulations showed that samples containing more furfuryl alcohol (namely: PFA, F/L 75/25 and F/L 50/50) hardened in shorter time: After 24 h at 100 °C and 1 h at 150 °C they were already completely cured. The formulations F/L 25/75 and the lactic acid in particular, were completely hardened after 285 h at 100 °C and finally 24 h at 150 °C. Therefore, the latter curing schedule was adopted to ensure curing completion for all samples. The so-made polymers underwent the complete characterization.

### 3.2. Bending Tests

The stabilized bioplastic blocks were tested for their bending resistance and the results are summarized in Table 1.

It can be observed that the formulations containing 25% and 50% of lactic acid showed a significantly higher MoR than the PFA while the formulation with 75% of lactic acid was considerably weaker. In order to better understand the behavior of the material, the stress/deflection graphic is presented in Figure 1.

The samples of PFA were characterized by a typical rigid behavior with sudden collapse of the material whereas the samples of PLA showed weak resistances and elastic behavior. The mixtures presented generally an intermediate behavior: With increasing amount of lactic acid the polymers became more elastic, visible in the decrease of the slope of the curves. Simultaneously the number and the intensity of micro-cracks increased. However, already small amount of FA in the formulation before curing (F/L 25/75) resulted in a sudden final rupture of the cured specimens. The increased elasticity of the material by adding small amount of lactic acid carried to improved mechanical performances because the presence of LA allowed a higher distribution of the strain all over the structure. In the formulation F/L 25/75 the amount of furanic skeleton was too weak and material collapsed after several consistent micro-cracks. These findings showed that the mixed formulations have intermediate properties: F/L 75/25 was more elastic than PFA and also more mechanically resistant, while F/L 25/75 was more rigid than PLA but also more resistant.

### 3.3. SEM Analysis

The morphology of the bioplastic was investigated via SEM and the major noticed differences are reported in Figure 2.

It can be observed that the PFA present several wrinkles (Figure 2a) due to the sudden rupture of the polymer layers (blue square). These wrinkles are linear and sharp (Figure 2b), similar patterns to glass-like materials. The PLA shown in Figure 2e,f presents a smoother surface with no wrinkles. On the side of Figure 2e, fibers can be observed suggesting a completely different microstructure. The image at higher magnification (2f) shows microfractures with pattern similar to landslide (red oval). The formulation F/L 50/50 presented intermediate features at µm scale. Less frequent but still linear cracks are observed in Figure 2c, while landslide microfractures are also observable at higher magnification (2d).

These observations may suggest that the structure of PFA in presence of lactic acid is less rigid because of the lower crosslinking degree.

### 3.4. Solubility Tests

The samples were grinded and immersed in water or chloroform for 1 h under continuous stirring and the solubility of the cured formulations is reported in Table 1. All polymers containing FA (PFA, F/L 75/25, F/L 50/50 and F/L 25/75) showed high water resistance with solubility rates lower than 2.5%, while the pure PLA formulation still contained some unreacted LA or small PLA oligomers carrying to lower water resistance (>9% solubility). This difference between the F/L 25/75 and PLA suggests that the presence of small amount of FA facilitate the polymerization of LA moieties. Also this test suggested that the presence of small amount of furanics produced higher property enhancement than the one expected from the blend.

Solubility in chloroform was much higher than the solubility in water. Neat PLA was completely soluble while neat PFA was insoluble. The intermediate blends were always less soluble than expected and more than 25% of the lactic fraction remained connected in the furanic network. This can be explained only by the presence of PFA–PLA copolymers or by the tight caging of the PLA in the PFA network.

### 3.5. Thermogravimetric Analysis

The TGAs traces of the mixed formulations presented an intermediate behavior between the one observed for pure PFA and pure PLA (Figure 3). All samples were thermally stable up to 300 °C followed by decomposition interval between 300 °C and 400 °C. While the pure PLA sample was completely degraded at 500 °C, the pure PFA sample resulted even in an incomplete degradation up to 1000 °C.

Interestingly, within the intermediate formulations (F/L 75/25, F/L 50/50 and F/L 25/75), the F/L 25/75 blend showed a significant difference: Its thermogram almost overlaps with the one of the F/L 50/50 strengthening the theory that the presence of a small amount of FA already modifies the thermal properties significantly. In a recent study, Nanni et al. observed that in PFA and polycaprolactone blends, the TGA traces resulted more similar to the ones of caprolactone involving complete degradation of the 50% blend before 450 °C [24]. Further, the observed behavior is in line with the bending and solubility tests and confirms that when small amounts of furfuryl alcohol are added the formulation markedly increase its furanic character.

### 3.6. Differential Scanning Calorimetry

The analysis of the DSC traces also highlights an intermediate behavior (Figure 4). Also in this test, the mixture F/L 25/75 presents higher furanic character than expected. This analysis confirms the thermal findings of the TG and further feeds the hypothesis of possible covalent interaction LA and FA when limited amount of FA are applied.

However, in order to identify the particular behavior of these formulations, spectroscopic analyses of the powders were performed.

### 3.7. FT-IR Spectroscopy

The spectra of the FT-IR analysis of the powders are presented in Figure 5.

The spectra of the mixed formulations generally show intermediate bands between the ones of polylactic acid and polyfurfuryl alcohol all over the spectral range. The most important region to be carefully observed is the C=O stretching were the vibration of the new ester may occur meaning between 1850 and 1650 cm^−1^. This region present major absorptions from the two homopolymers: The PFA absorb from 1790 until 1640 cm^−1^ because the ketons of the cross-linked ring-opened structure of PFA absorb at 1710 cm^−1^ two main shoulders at 1770 cm^−1^ (small) and 1680 cm^−1^ (medium) due to terminal-lactones and to the C=O vicinal to C=C in the linear arrangement also contribute to this band; the PLA has a very intense broad band at 1750 cm^−1^ due to the C=O stretching of the ester. This means that if a new esterification occurs, the new signals would easily overlap with the absorptions of the homopolymers and hence their detection would be complicated.

### 3.8. Solid State ^13^C–NMR

Solid state ^13^C–NMR was also performed and the spectra obtained are summarized in Figure 6.

With this technique the spectra also show a clear trend for every signal depending on the relative amount of the components in the formulation. Clearly the signal in the region of 108 ppm, corresponding to the C3 and C4 of the furfuryl alcohol and those at 151 ppm (C2 and C5) increase with increasing FA concentration. Also the methylene bridge signal at 27 ppm shows this trend. On the other hand, the signals of the carboxyl group at 172 ppm, the CH-ether at 68 ppm and the methyl group at 17 ppm of the PLA decreases proportionally. However, eventual new signals due to possible crosslinking would have occurred at around 175–170 ppm in case of ester bond between lactic acid and furfuryl alcohol moieties. Indeed, a small shoulder signal in this region occurs at around 175 ppm (top left frame) and we believe that this could be due to a contained ester formation between the two moieties. The easiest example of possible esterification pathway is hypothesized in Figure 7.

Summarizing, we can state that the great majority of the molecules of FA and LA homopolymerize to PFA and PLA but there are several clues that suggest that a minor number of hetero-esters may occur giving place to a block copolymer which assumes a furanic character also when contained amount of FA is applied (F/L 25/75).

## 4. Conclusions

In this study, macroscopic homogeneous furfuryl alcohol and lactic acid blends were successfully synthesized and their characterization showed that:-Bending resistance of the formulations was enhanced and as soon as small amounts of FA were added, the formulation became rigid, producing sudden ruptures.-The SEM micrographs of the mixed formulations showed patterns of both homopolymers.-Higher furanic character then expected was observed for the formulations F/L 25/75 when exposed to water solubility test, TG and DSC analyses.-Sensibly higher resistance to chloroform then expected (>25%) was observed for all the mixed formulations.-No evidences of co-polymerization could be observed in FT-IR, while a small shoulder signal at 175 ppm in the ^13^C–NMR spectra of the mixed formulations reinforced the thesis of possible copolymerization occurrence.

Is this evidence enough to claim the synthesis of a new copolymer? We do not know yet, but in any case, we are convinced that the two homo-polymers are the broad majority and that a grafting process may occur when limited amount of FA are applied.

## Figures and Tables

**Figure 1 polymers-11-01533-f001:**
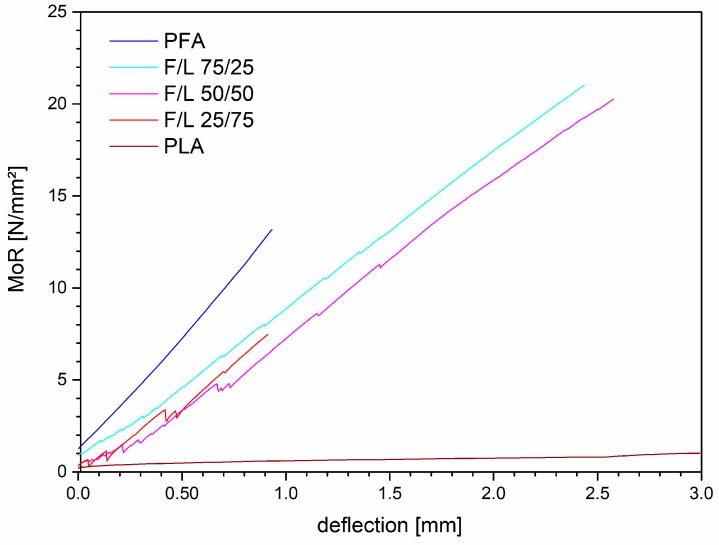
Bending strength of PFA, PLA and their mixtures.

**Figure 2 polymers-11-01533-f002:**
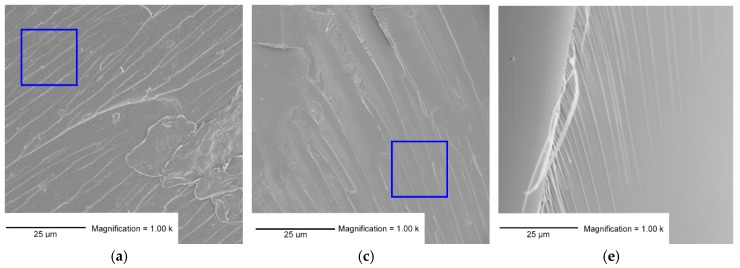
SEM micrographs of PFA (**a**,**b**), F/L 50/50 (**c**,**d**) and PLA (**e**,**f**); magnification 1000 (top) and 15,000 (bottom).

**Figure 3 polymers-11-01533-f003:**
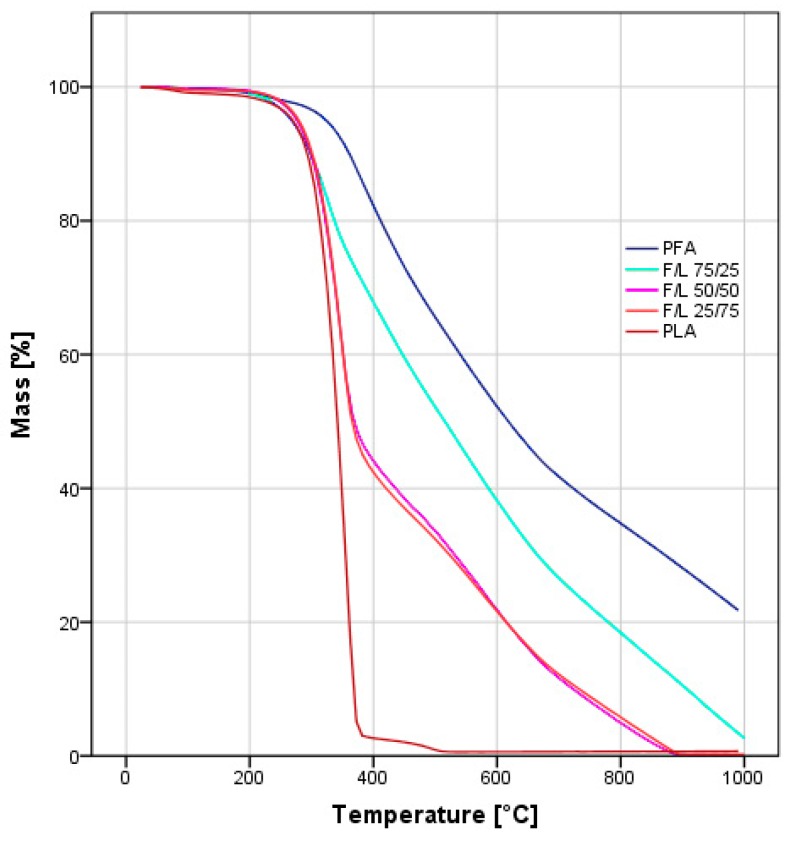
TGA analysis under synthetic air of the samples from pure PFA and PLA as well as the mixed formulations with ratios F/L 75/25, F/L 50/50, F/L 25/75.

**Figure 4 polymers-11-01533-f004:**
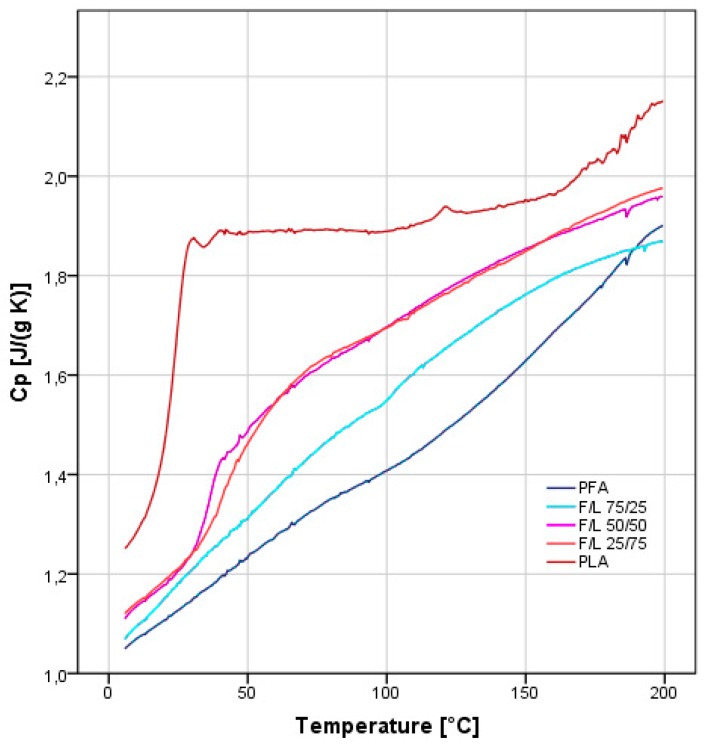
Differential scanning calorimetry of the 5 samples: PFA, F/L 75/25, F/L 50/50, F/L 25/75 and PLA.

**Figure 5 polymers-11-01533-f005:**
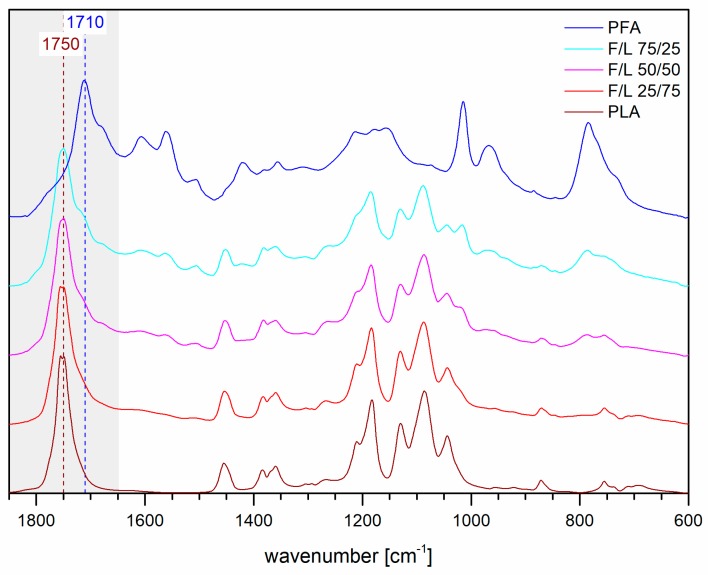
FT-IR spectra of PLA, PFA and the three mixed formulations.

**Figure 6 polymers-11-01533-f006:**
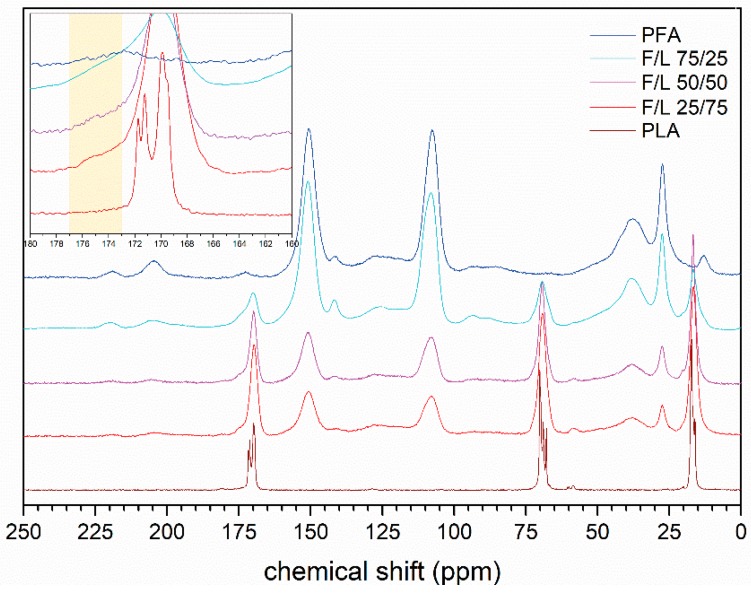
Solid state ^13^C–NMR spectra of PLA, PFA and three mixed formulations.

**Figure 7 polymers-11-01533-f007:**
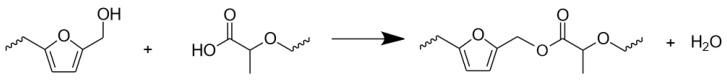
Possible copolymerization reaction between FA and LA moieties.

**Table 1 polymers-11-01533-t001:** Formulation of the biopolymers in % by weight, bending resistance & water solubility.

Label	Furfuryl Alcohol (%)	Lactic Acid (%)	Bending Resistance (MoR) ^1^ (N/mm^2^)	Solubility in Water (%) ^1^	Solubility in Chloroform (%) ^1^
PFA^−2^	100	0	13.7 (5.6)	2.36 (1.08)	1.23 (0.20)
F/L 75/25	75	25	28.1 (6.3)	1.55 (0.89)	18.46 (0.21)
F/L 50/50	50	50	25.4 (1.9)	2.05 (1.04)	31.15 (0.83)
F/L 25/75	25	75	7.2 (0.3)	2.33 (1.44)	55.45 (3.20)
PLA	0	100	Elastic	9.12 (13.42)	98.29 (2.37)

^1^ Standard deviations are reported in brackets. ^2^ 0.2 mL of sulfuric acid were added to start the polymerization.

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
