# Peer review of "Furfuryl Alcohol and Lactic Acid Blends: Homo- or Co-Polymerization?"

_polymers, 2019, doi:10.3390/polym11101533_

Round 1

Reviewer 1 Report

The manuscript entitled ' Furfuryl alcohol and lactic acid blends: Homo- or co-polymerization? ' by Lukas Sommerauer and Jakub Grzybek is well-organized in which authors provided an interesting verification that most of furfuryl alcohol and lactic acid blends undergo homopolymerization and a small number of them may undergo copolymerization. This work used several methods to prove differences with the addition of FA, and explored chemical mechanism of FA and LA blends. Therefore, this paper should be published in Polymers after major revision.

There are many punctuation and grammatical errors throughout the paper. The English should be polished. In the introduction, there must be other copolymers like FA and LA blends, it’s better to introduce some different copolymers and their application. The caption in SEM images is not clear. The author can mark directly in the images. The DSC of PFA,F/L 75/25, F/L 50/50, F/L 25/75 and PLA should be analyzed one by one. In figure 5, the colors of PLA, PFA and three mixed formulations are too similar to be distinguished. It is better to provide the solid state 1H-NMR spectroscopy of PFA, PLA and the mixed formulations to reinforce the thesis of possible copolymerization occurrence. Whether the molecular weight of PFA and PLA affects the physicochemical property of materials? Whether PFA and PLA with the same mixing ratio and different degree of polymerization affect the properties of the materials? Whether copolymerization has certain advantages compare with direct blending of two polymers? Some important papers about polymers for biomedical applications have been published in recent years, please update the reference section, for example: Journal of Materials Chemistry B 2019, 7, 709–729. Advanced Materials, 2018, 1803217.

Author Response

The manuscript entitled ' Furfuryl alcohol and lactic acid blends: Homo- or co-polymerization? ' by Lukas Sommerauer and Jakub Grzybek is well-organized in which authors provided an interesting verification that most of furfuryl alcohol and lactic acid blends undergo homopolymerization and a small number of them may undergo copolymerization. This work used several methods to prove differences with the addition of FA, and explored chemical mechanism of FA and LA blends. Therefore, this paper should be published in Polymers after major revision.

Comment: Thank you very much for the time you spent in the review of our paper and for the remark done to improve our previous version.

There are many punctuation and grammatical errors throughout the paper. The English should be polished.

Answer: We have also observed several spelling and grammar and we corrected them all (we wish).

In the introduction, there must be other copolymers like FA and LA blends, it’s better to introduce some different copolymers and their application.

Answer:  We have extended this part and we have presented literature studies involving PFA and PLA alone as well as PFA copolymers (with natural aromatic compounds)  and PLA copolymers (with PEG and PCL).

The caption in SEM images is not clear. The author can mark directly in the images.

Answer:  Caption of the SEM images was modified to clarify the connection of text and image

The DSC of PFA,F/L 75/25, F/L 50/50, F/L 25/75 and PLA should be analyzed one by one.

Answer: We have structured the DoE of this paper in order to observe differences from the trend by producing mixtures of the two monomers. We believe that describing the single behavior in the DSC of every blend, would mean that we also have to describe the same for the TG, for the 13C-NMR and for the FT-IR. To my sight, this will increase the dimension of the paper, but these information will not help in focusing effectively in the objective of this research which is resumed in the title of the paper. Unless absolutely necessary, I would prefer not modifying the structure of the paper.

In figure 5, the colors of PLA, PFA and three mixed formulations are too similar to be distinguished.

Answer: The figure has been reworked, it should now be easier to distinguish

It is better to provide the solid state 1H-NMR spectroscopy of PFA, PLA and the mixed formulations to reinforce the thesis of possible copolymerization occurrence.

Answer: Thank you for the valuable suggestions. When we assessed the NMR analysis, we decided that major changes would have been visible with 13C because the surrounding of the Hydrogens would have been even more similar (in case of effective copolymerization). Being these formulations constituted of macromolecules, the H-signals would be more band-like and we suspected that they would be less effective than the 13C-NMR.

Whether the molecular weight of PFA and PLA affects the physicochemical property of materials?

Answer: Thank you for the nice input. Unfortunately, the study of the molecular weight of these compound is very complex because PFA is not soluble in DMSO and therefore GPC will not be possible. This test could only give us information on the MW of the soluble PLA part. Also in this case we do not have access to this facilities in our labs. Taking this hint, we have decided to repeat the solubility test also with chloroform (were PLA is soluble and PFA is not) in order to understand if part of the PLA is connected with the PFA. The results were very interesting and they support the copolymerization theory presented.

Whether PFA and PLA with the same mixing ratio and different degree of polymerization affect the properties of the materials?

Answer:  Thank you for this interesting observation. The degree of polymerization of PFA is hard to control. In future studies we would like to increase the copolymerization between the PLA and PFA. Maybe we could accelerate the PLA polymerization by adding catalysts, or decreasing the PFA by keeping the reaction in more alkaline conditions.  

Whether copolymerization has certain advantages compare with direct blending of two polymers?

Answer: Thanks for the suggestion. The copolymerization is the only way to produce a stable blend. PFA is thermosetting and cannot be melted with PLA to produce an homogeneous product. So the direct blending would be a PLA matrix with PFA powders dispersed inside.

Some important papers about polymers for biomedical applications have been published in recent years, please update the reference section, for example: Journal of Materials Chemistry B 2019, 7, 709–729. Advanced Materials, 2018, 1803217.

Answer: Reference 1 was added in the introduction section, reference 2 could not be found/ or it does not really fit with the topic of this paper.

Reviewer 2 Report

Dear authors

I have reviewed the paper entitled ‘Furfuryl alcohol and lactic acid blends: Homo- or co-polymerization?’. The work reported is interesting and the paper is well written. The paper can be published after minor corrections.

L39: please add a value (or range) for ‘high temperature

Table 1: PLA, what do you mean by No for the bending resistance. Add a comment in the legend  of the table.

Figure 6: use more contrasted colors for the different polymers to improve the reading

L245: it is difficult to identify the shoulder on Fig 6. Please add on Fig6 (top image) for each curve where is exactly the shoulder.

Best regards

Author Response

Dear authors

I have reviewed the paper entitled ‘Furfuryl alcohol and lactic acid blends: Homo- or co-polymerization?’. The work reported is interesting and the paper is well written. The paper can be published after minor corrections.

Comment: Thank you very much for your review, it is never easy to find time for reviewing.

L39: please add a value (or range) for ‘high temperature

Answer: Temperature up to which PFA is mostly stable was added including a reference

Table 1: PLA, what do you mean by No for the bending resistance. Add a comment in the legend of the table.

Answer: Changed “no” to “elastic” as there was no reasonable result due to the elastic behaviour of our PLA.

Figure 6: use more contrasted colors for the different polymers to improve the reading L245: it is difficult to identify the shoulder on Fig 6. Please add on Fig6 (top image) for each curve where is exactly the shoulder.

Answer: Region in which the shoulder occurs is highlighted in yellow

Reviewer 3 Report

The submitted paper has been written in a way that makes it inadequate to qualify as scientific. Most of the claims made by the authors are unsubstantiated or simply unrealistic. The style of writing is not acceptable bearing in mind the Journal guidelines. The examples below clearly prove that the paper is still a work in progress, with only initial research and largely unfounded results.

The Abstract:

“Typically, FA homopolymerize exothermically in acid environment producing  inhomogeneous porous materials, but recent studies have shown that this reaction can be controlled”

Such information should be included in the Introduction section of the paper with appropriate references to specific publications in which the reaction in question was discussed.

- On reading the Abstract it is not possible to establish if Authors prepared a copolymer or blends. There is a significant difference between a copolymer and a blend and a clear distinction should be made..

- The Authors claim that “The mechanical tests have shown that the blend containing small amount of FA were rigid” Bearing in mind that FA and LA are liquids it is unreasonable and unjustified to claim that the blend was (not were) rigid since this term clearly indicates a solid not a liquid.

 - Keywords have been inadequately selected (for example:  sustainable, formulation)

-In my opinion the conditions described in the section “Synthesis of PFA-PLA polymers“ are inappropriate for the polymerization process. bearing in mind that there is no indication of a catalyst and additional pressure being applied, it is unreasonable  to claim that polymers were obtained. DSC results clearly prove that PLA was not obtained.

-There is no explanation as to why the “Solubility test” was performed in water? Polymers that are the object of the study are generally not soluble in water.

-In the case of “Thermo-gravimetric analysis” insoluble fraction was analyzed. There is no indication whether the in the case of the other analyses also the insoluble fraction was taken into account or not.

- Bearing in mind the overall angle of the paper, there is no point in presenting the SEM results.

- The C13 NMR technique, which is the most important in the case of structural study of the obtained materials, has not been sufficiently described. There is no connection between individual signals and particular carbon groups. A structure pertaining to specific materials with particular signals assigned assigned to those materials should have been included. Moreover, in the case of scientific research a “belief” (page 9 line 246) is not enough to substantiate  incomplete experiments.

In the conclusions Authors write: “Are these evidences enough to claim the synthesis of a new copolymer? We do not know yet,….”

One of the final statements included it the paper is a clear indication that this work is not ready for publication.

Author Response

The submitted paper has been written in a way that makes it inadequate to qualify as scientific. Most of the claims made by the authors are unsubstantiated or simply unrealistic. The style of writing is not acceptable bearing in mind the Journal guidelines. The examples below clearly prove that the paper is still a work in progress, with only initial research and largely unfounded results.

Comment: I thank the reviewer for the time spent in reviewing the paper. I have to admit that after several years of carreer this is possibly the worst commentary I have ever received. I believe the reviewer might be emotionally involved in the topic or with the authors. As corresponding author of the paper I am ready to defend what we stated in the paper and to clarify the meaning of every statement when I will be contacted. I believe this investigation is complete, but I agree also that further studies can be done in order to optimize the copolymerization of the precursors.

The Abstract:

“Typically, FA homopolymerize exothermically in acid environment producing inhomogeneous porous materials, but recent studies have shown that this reaction can be controlled”

Such information should be included in the Introduction section of the paper with appropriate references to specific publications in which the reaction in question was discussed.

Answer: These information were abundantly reported in line 44 to 50.

- On reading the Abstract it is not possible to establish if Authors prepared a copolymer or blends. There is a significant difference between a copolymer and a blend and a clear distinction should be made.

Answer: The last sentence of the abstract clearly states that there is evidence of a possible copolymer. Of course we know the differences between copolymer and blend, indeed we spent the whole research described in this article to try to understand if covalent bond between LA and FA  occurred or not.

- The Authors claim that “The mechanical tests have shown that the blend containing small amount of FA were rigid” Bearing in mind that FA and LA are liquids it is unreasonable and unjustified to claim that the blend was (not were) rigid since this term clearly indicates a solid not a liquid.

Answer: As all tests, bending resistance was measured at the cured sample. PFA and PLA are solid at room temperature after polymerization. We guess that the reviewer means that we could have written PFA instead of FA in this blend. But we cannot guarantee of having PFA at this stage, we can only state that the formulations containing small amount of FA before polymerization were rigid. The sentence as has been changed accordingly.

 - Keywords have been inadequately selected (for example:  sustainable, formulation)

Answer: Keywords were modified

-In my opinion the conditions described in the section “Synthesis of PFA-PLA polymers“ are inappropriate for the polymerization process. bearing in mind that there is no indication of a catalyst and additional pressure being applied, it is unreasonable to claim that polymers were obtained.

Answer:  PLA can be prepared without catalyst or additional pressure. PFA need a catalyst (slightly acid conditions may be enough when enough energy is supplied. The catalyst was lactic acid acting as catalyst and precursor. In case of neat PFA, sulfuric acid was added. That was clearly presented in the experimental section.

DSC results clearly prove that PLA was not obtained.

Answer: The DSC curve of PLA clearly shows two peaks at 60 and 150 °C which are characteristic for PLA as proven by other research.

-There is no explanation as to why the “Solubility test” was performed in water? Polymers that are the object of the study are generally not soluble in water.

Answer: Thanks for the input. This was a really interesting point. Results for the solubility in chloroform (in which PLA is soluble) were added and it was clarified in the material and methods part that solubility in water was mainly done to remove unreacted precursor.

-In the case of “Thermo-gravimetric analysis” insoluble fraction was analyzed. There is no indication whether the in the case of the other analyses also the insoluble fraction was taken into account or not.

Answer: It was clarified that for all tests (except bending resistance) the water leached powder was used

- Bearing in mind the overall angle of the paper, there is no point in presenting the SEM results.

Answer: Thanks for your observation. We have thought about removing the SEM results and we decided to present them anyways because it was nice to observe the intermediate rupture behaviour of  the mixed formulations, which presented both crack and land-slide patterns. I believe that this carry some further information and hence we decided to present it. We agree that this does not really help in understanding if the copolymerization between LA and FA occurred or not.

- The C13 NMR technique, which is the most important in the case of structural study of the obtained materials, has not been sufficiently described. There is no connection between individual signals and particular carbon groups. A structure pertaining to specific materials with particular signals assigned to those materials should have been included.

Answer: Thank you for the input. We have extended the 13C-NMR part accordingly.

Moreover, in the case of scientific research a “belief” (page 9 line 246) is not enough to substantiate incomplete experiments.

Answer: Well, a theory is true until is confuted. At present we believe we have partially copolymerized LA and FA but someone else in future may prove that we were wrong. Indeed, we used an extensive number of techniques but still, it was hard to detect unconfutable polymerization evidences. In future studies we will try to extend the copolymerization so that we wish we can have some, even more concrete, evidence of polymerization. We believe that the research in this direction is definitely under progress, but the obtained results are already concrete enough for being presented to the scientific community.

Round 2

Reviewer 1 Report

The authors generally addressed the comments from the reviewers, and the manuscript is recommended for publication.